# Biomarkers of Relapse in Cocaine Use Disorder: A Narrative Review

**DOI:** 10.3390/brainsci12081013

**Published:** 2022-07-30

**Authors:** Margaux Poireau, Thomas Milpied, Angéline Maillard, Christine Delmaire, Emmanuelle Volle, Frank Bellivier, Romain Icick, Julien Azuar, Cynthia Marie-Claire, Vanessa Bloch, Florence Vorspan

**Affiliations:** 1Département de Pharmacie, Université de Paris Cité, Inserm UMR-S 1144, Optimisation Thérapeutique en Neurospsychopharmacologie, OTeN, F-75006 Paris, France; margaux.poireau@aphp.fr (M.P.); thomas.milpied@aphp.fr (T.M.); angeline.maillard@aphp.fr (A.M.); christine.chd@gmail.com (C.D.); frank.bellivier@inserm.fr (F.B.); romain.icick@aphp.fr (R.I.); julien.azuar@aphp.fr (J.A.); cynthia.marie-claire@inserm.fr (C.M.-C.); vanessa.bloch@aphp.fr (V.B.); 2Département de Psychiatrie et Médecine Addictologique, Hôpital Fernand Widal, Assistance Publique–Hôpitaux de Paris Nord, 75010 Paris, France; 3Fédération Hospitalo-Universitaire Network of Research in Substance Use Disorders, 75010 Paris, France; 4Service de Neuroradiologie, Fondation Ophtalmologique Rothschild, 75019 Paris, France; 5FrontLab at Paris Brain Institute (ICM), Sorbonne University, Inserm, Centre National de la Recherche Scientifique, 75013 Paris, France; emmavolle@gmail.com; 6Assistance Publique–Hôpitaux de Paris Nord, Service de Pharmacie, Hôpital Fernand Widal, 75010 Paris, France

**Keywords:** biomarkers, treatment response, relapse, cocaine use disorder

## Abstract

Introduction: Cocaine use disorder is a chronic disease with severe consequences and a high relapse rate. There is a critical need to explore the factors influencing relapse in order to achieve more efficient treatment outcomes. Furthermore, there is a great need for easy-to-measure, repeatable, and valid biomarkers that can predict treatment response or relapse. Methods: We reviewed the available literature on the Pubmed database concerning the biomarkers associated with relapse in CUD, including central nervous system-derived, genetic, immune, oxidative stress, and “other” biomarkers. Results: Fifty-one articles were included in our analysis. Twenty-five imaging brain anatomic and function assessment studies, mostly using fMRI, examined the role of several structures such as the striatum activity in abstinence prediction. There were fewer studies assessing the use of neuropsychological factors, neurotrophins, or genetic/genomic factors, immune system, or oxidative stress measures to predict abstinence. Conclusion: Several biomarkers have been shown to have predictive value. Prospective studies using combined multimodal assessments are now warranted.

## 1. Introduction

Cocaine is a psychoactive substance originating from coca leaves (Erythroxylum coca plant). The leaves are soaked in solvents and transformed into a paste and then into powder to obtain a drug composed of cocaine hydrochloride. In its most common powder form, cocaine can be snorted or injected, but it can also be inhaled when transformed into crack-cocaine [1]. By blocking specific transporters (DAT, SERT, NET), its consumption causes the accumulation of neurotransmitters, such as dopamine, serotonin, and noradrenaline, in the synaptic cleft. It results in a stimulating feeling of energy, disinhibition, euphoria, and grandiosity. 

In 2019, roughly 0.4% of the world adult population aged 15–64 years had used cocaine. Cocaine use is mostly heterogeneous, implanted in many different social environments, and has increased in recent years [2]. 

Cocaine use disorder (CUD) is classified using the Diagnostic and Statistical Manual of Mental Disorders (DSM-5). Subjects with CUD are unable to stop their cocaine use, despite multiple consequences, which may lead to dependence. CUD is thus a major public health problem. This disorder is a multifactorial pathology with consequences that can be somatic, psychiatric, cognitive, and social. Somatic complications include, among others, cardiovascular, neurological, respiratory, otolaryngological, and dermatological pathologies or infectious complications (in particular caused by risky consumption behaviors) [3,4]. Regarding the psychiatric consequences, patients are likely to develop psychotic disorders or behavioral disturbances, and to have depressive episodes or even suicidal ideas [5,6,7,8].

Wagner and Anthony determined that after the first use of cocaine and over a 10-year follow-up period, dependency (the worst stage of CUD) developed in 15% to 16% of cocaine users [9]. Cocaine-dependent subjects (according to the DSM IV definition), also described as subjects with severe CUD (at least six criteria according to the new DSM 5 definition), are the ones who seek treatment. Patients with cocaine dependence are characterized by a loss of control over consumption, and a feeling of craving when they do not use the drug. Craving is central to cocaine dependence. It involves an intense desire or even an imperious need to consume cocaine, as well as physical symptoms of arousal that mimic the anticipated physical signs of cocaine intake [10]. Craving appears spontaneously or can be triggered by an external stimulus such as the cocaine itself or the money to buy it, the people associated with cocaine use, or consumption paraphernalia. Users feel a very strong need to consume cocaine that is very difficult to fight [11]. 

However, even when these patients seek treatment, the response rate to available care is still low. The maintenance rate of abstinence of cocaine-dependent patients after hospital withdrawal was shown to be about 25% after one year in a US study [12]. Despite a large number of current clinical trials, there is no identified substitutive pharmacological treatment for CUD, and the lack of efficient treatment leads to a high relapse rate for CUD [13]. Psychotherapies remain the treatments of choice for cocaine dependency, particularly behavioral and cognitive approaches that are employed during outpatient or inpatient cessation attempts. The most severely affected patients require inpatient treatment to allow them to receive care when their environment itself is a cue that triggers craving and cocaine use. In this context of low treatment responses, there is no biomarker to predict individual responses to routine care. This is why there is a great need to increase knowledge on CUD, both in terms of the pathophysiology of this disorder from instatement to relapse, and in terms of specific and valid predictive biomarkers of relapse. 

Some clinical factors may be associated with the severity of CUD and thus are associated with relapse after cocaine cessation. The most well-known clinical predictors of relapse are the frequency and quantity of cocaine use per month [14,15,16], the number of CUD DSM 5 criteria met [17], craving scores [18], years of education [19,20], and the impulsivity of the user [21]. Moreover, greater severity of CUD has been associated with a greater vulnerability to relapse as well as a shorter abstinence time [22].

In addition to clinical factors, some studies have also focused on imaging, molecular, and cellular biological biomarkers, aiming to link them to the severity of CUD from chronic drug use as well as the disease trajectory, and adherence to care [23]. 

Indeed, when cocaine addiction is instated, the repeated use of cocaine has already altered brain homeostasis, particularly in the reward system. Signs of withdrawal and craving accompany the absence of consumption. Predictive biomarkers of relapse could indicate this new cerebral homeostasis. Given the difficult access to the living brain, biomarkers of psychiatric or addictive disorders should ideally be peripheral, easy to measure, repeatable, a valid reflection of pathological brain function, sensitive to change, predictive of either the staging or time course of the disorder, or predictive of the therapeutic response (pharmacotherapeutic or psychotherapeutic) [24,25].

Considering the new state of homeostasis generated by cocaine addiction, and in agreement with the definition of the ideal biomarker in addictive disorders, we searched for articles on biomarkers in different domains related to this change in homeostasis.

In particular, these biomarkers belong to four domains: anatomic/functional biomarkers of the central nervous system (CNS), genetic/genomic biomarkers, biomarkers of the immune system/hypothalamo-hypophyso-adrenal axis/oxidative stress, and others. 

## 2. Methods

For this narrative review, original research papers and literature reviews were identified from the Pubmed database, including articles published between January 2000 and September 2021. The keywords, which were used in different combinations, were: Cocaine, Biomarkers, Abstinence, Relapse, Treatment Response, Humans, and either Genetic, BDNF, Neuroimaging, or Cytokines, in accordance with our hypothesis. This literature search identified 311 articles. Titles and abstracts were screened by two independent readers (M.P and T.M.) using predefined criteria. Exclusion criteria were as follows: all studies published before 2000, pre-clinical studies on experimental models, prenatal and post-mortem studies, studies not assessing CUD, and studies not assessing relapse or closely related outcomes. The first screening left 105 studies, which then underwent full reading with the same exclusion criteria. The final selection identified 35 articles that matched our search criteria. After reading the references cited in these articles, we added 16 articles that matched our inclusion criteria. Thus, a total of 51 articles were included in this analysis. 

Following our initial hypothesis, the results are presented in four domains: anatomic/functional biomarkers of the CNS, genetic/genomic biomarkers, biomarkers of the immune system/hypothalo-hypophyso-adrenal axis/oxidative stress, and “other” biomarkers. As some articles assessed several biomarker types at the same time, the sum of articles in the different categories is greater to the total of articles analyzed, leading to a total of 57 articles in the seven categories. Articles testing the association between clinical factors and relapse are presented in the introduction.

## 3. Results

After reading and analyzing the details of the 51 articles testing the association of several biomarkers with relapse in CUD, we classified them into four domains: anatomic/functional biomarkers of the CNS, genetic/genomic biomarkers, biomarkers of the immune system/hypothalo-hypophyso-adrenal axis/oxidative stress, and “other” biomarkers. The anatomic/functional biomarkers of the CNS domain were themselves divided into three categories for easier reading: brain anatomic and function assessment, neuropsychological and cognitive challenges, and peripheral markers of CNS function.

A flow-chart of the article selection process is presented in Figure 1.

### 3.1. Central Nervous System

#### 3.1.1. Brain Anatomic and Function Assessment

Twenty-five articles were relevant to the study of CNS biomarkers in relation to relapse. We divided these articles into several thematic categories: brain anatomic and function assessment articles, neuropsychological and cognitive testing articles, and articles studying neurotrophins, neurotransmitters, and brain metabolites.

Among the neuroimaging studies, most studies compared subjects suffering from cocaine dependency, and only a few explored the factors influencing relapse and abstinence among cocaine-dependent subjects (see Table 1).

We nevertheless identified thirteen studies exploring relapse with task-based functional imaging (fMRI). Most studied brain activity or connectivity during either cue-induced craving tasks, or direct or indirect measures of inhibition. Thus, they were designed to assess the reward system in situations relevant to CUD or relapse prevention.

The most consistent result was the functional role of the striatum in CUD severity and abstinence. The results particularly highlighted the role of the ventral striatum and, therefore, the nucleus accumbens (NAcc), which is considered a critical node for motivation [48]. An increase in activity in this area was associated with a decrease in abstinence in two prospective studies involving treatment-seeking cocaine-dependent patients [26,32]. Furthermore, increased cue-induced cravings have been associated with increased right ventral striatum activity [18].

The reviewed studies were less consistent regarding another region of the reward system, the prefrontal cortex (PFC), which is involved in positive reinforcement. Better maintenance of abstinence was associated with increased activity of the PFC in a cross-sectional study on abstinent CUD patients [29], and with decreased activity in the dorsolateral PFC in a prospective study [27]. Finally, greater cocaine consumption in the previous month was associated with increased activity in the ventromedial PFC (vmPFC) and the thalamus in a cross-sectional study of fifty-four cocaine-dependent patients [36].

Regarding the thalamus, inconsistent results were observed. Decreased activity has been associated with greater abstinence maintenance [30,32], while another study associated decreased thalamus activity with worse abstinence maintenance [33]. 

Apart from task-based fMRI studies, only a few imaging techniques have been used to describe biomarkers of relapse in CUD.

Five studies used the resting-state imaging technique, which evaluates the intrinsic functional connectivity (rsFC) between brain regions. Indeed, two prospective studies identified a stronger rsFC between several structures related to a higher relapse rate (including higher connectivity between the PCC and the precuneus, or the posterior cingulate cortex and the NAcc) [37,38]. Gu et al. also associated a weaker rsFC between the ventral tegmental area and the NAcc and other structures with more years of cocaine use [39]. Finally, McHugh et al. identified several structures involved in relapse with fMRI performed in the final week of a residential treatment episode. A decreased rsFC in the basolateral amygdala, the lingual gyrus, and the cuneus was associated with abstinence maintenance, whereas a decreased rsFC in the left corticomedial amygdala, the vmPFC, and the rostral anterior cingulate cortex was associated with relapse [19]. Additionally, another recent study associated higher connectivity between several prefrontal cortex regions of interest with a better treatment outcome, i.e., a lower likelihood of relapse [40].

Only one study examined the role of structural connectivity in CUD. This study used diffusion tensor imaging (DTI) and showed a relationship between a higher white matter (WM) integrity (i.e., fractional anisotropy values) in several structures (such as the frontal, temporal, parietal, and occipital lobes) and better abstinence maintenance [41].

According to the few studies that used positron emission tomography (PET-SCAN) imaging, better dopamine signaling was associated with better abstinence maintenance [44]. 

Additionally, higher levels of creatine, choline-containing metabolites, and myo-Inositol were associated with greater abstinence in patients with polysubstance use using high field brain magnetic resonance spectroscopy SPECT [42]. Furthermore, greater cocaine consumption has been associated with lower GABA and N-acetylaspartate levels [43].

Morphometry studies based on anatomical MRI associated a smaller grey matter (GM) volume in the cingulate cortex and the bilateral superior frontal gyri with a longer cocaine use duration (years) in CUD patients [45]. A larger white matter volume in the hippocampus was associated with a shorter duration of continuous abstinence [46].

We only identified one study that used the late positive potential (LPP) recorded on EEG as an objective measure of attention and salience of visual stimuli in a cue-induced craving challenge. This study compared different groups of cocaine users at different stages of abstinence, but no prospective follow-up was performed to observe relapse. Still, this technique could be the first step toward EEG-predicted relapse [47].

#### 3.1.2. Neuropsychological and Cognitive Challenges

To assess neuropsychological biomarkers as predictors of abstinence, relapse, or treatment response in CUD, most studies have used cue-induced challenges. The neuropsychological dimensions assessed in the challenges were craving, impulsivity, or inhibitory capacity. The results are detailed in Table 2. 

Four studies examined the predictive value of the Stroop test for assessing attentional bias. They showed that an increase in attentional bias toward cocaine in a cocaine-word Stroop task was associated with treatment dropout [49]. Furthermore, an increased interference effect in the Comalli–Kaplan version of the Stroop task has also been associated with treatment dropout [51]. Additionally, the interference effect has been positively correlated with the number of self-reported days of abstinence, and negatively correlated with positive urine samples in treatment-seeking dependent patients [32]. Finally, Brewer et al. found that the interference effect correlates positively with treatment retention [27].

Other authors used several memory and mental flexibility assessments to examine the relationship with relapse. Fox et al., using the Rey Auditory Verbal Test (RAVLT, which evaluates verbal learning and memory), observed that lower memory scores were associated with a higher risk of relapse [50]. Furthermore, a higher incidence of relapse was associated with poorer performance on the Wisconsin Card Sorting Test (WCST) (which evaluates perseverance and the ability to adapt the cognitive strategy) [20].

#### 3.1.3. Peripheral Biomarkers of CNS Functioning

There has been a trend toward the development of plasmatic dosing of several proteins that are synthetized in the brain. These proteins are candidates for peripheral biomarkers of CNS function. 

One of the most frequently studied biomarkers is brain-derived neurotrophic factor (BDNF), also a candidate biomarker for staging and treatment responses in several psychiatric disorders, especially mood disorders and suicide [52]. It was also found to be one of the most frequently studied in CUD research in this literature review. The results are detailed in Table 3.

Studies investigating the predictive value of BDNF in CUD relapse have revealed divergent results. While some studies have observed that a longer duration of abstinence is correlated with higher levels of plasmatic BDNF [54,57], others have reported the opposite result [15,53]. In a prospective study conducted during a three-week inpatient cocaine cessation period, Sordi and collaborators observed that patients had significantly lower BDNF plasmatic levels at entry than a healthy control group, but their BDNF levels rose during the inpatient stay. There was also a correlation between the severity of CUD and the BDNF level at discharge [58]. 

A study assessing CNS-derived plasmatic metabolites showed that more severe CUD was associated with higher n-methylserotonin levels and lower xanthine levels [56]. 

### 3.2. Genetic and Genomic Biomarkers

Despite the vast genetic literature produced over the past twenty years, including case-control research describing genetic risk factors for developing CUD using either candidate genes or hypothesis-free genome wide association studies [59], only a few studies have assessed the genetic risk of CUD relapse. In this category, we identified two relevant articles describing a significant association between genetic polymorphisms of candidate genes and relapse (for details see Table 4). One of them also included a combined neuropsychological assessment [49].

The development of genomic tools to assess gene expression and its links to relapse or treatment response in CUD is in the early stage. However, we identified one article showing a positive association of serum miR-181 expression during inpatient cocaine cessation with stronger dependency and greater cognitive and depressive withdrawal symptoms [61], but not with the prediction of relapse. 

### 3.3. Immune System and Oxidative Stress Biomarkers

Over the past 20 years, the body of literature focused on immune and inflammatory processes associated with brain diseases has been growing. This is especially true in terms of psychiatric disorders, while research is only just beginning for addictive disorders. Immune cells play a role in the development of several psychiatric diseases [62], and patients with chronic psychiatric disorders experience chronic low-grade inflammation processes, leading to the development of plasmatic predictive biomarkers [63]. More specifically, regarding CUD, nine studies have assessed the potential predictive roles of the immune system and oxidative stress peripheral biomarkers in abstinence. They are presented in Table 5.

Most of this research investigated how the interleukin proteins could be associated with CUD severity or relapse, mainly through cross-sectional studies. 

It was shown that a higher plasmatic IL-6 level is associated with lower cognitive flexibility, assessed by the Wisconsin CARD Sorting Test (itself previously associated with the risk of relapse) [67]. In a more recent study, a longer duration of cocaine use was associated with lower IL-9 and IL-20 levels [70]. Concerning IL-1β, lower plasmatic levels were observed in CUD patients with at least two weeks of abstinence compared with healthy controls [64]. 

Regarding cytokines with chemoattractant properties, also known as chemokines, a decrease in monocyte chemoattractant protein 1 (MCP-1) and stromal cell-derived factor 1 (SDF-1) levels have been observed in CUD patients assessed after at least 2 weeks of abstinence compared with healthy controls. Surprisingly, in the same cross-sectional study, higher levels of fractalkine and SDF-1 were observed in the subgroup of patients with more severe CUD [64].

Regarding the other cytokines, the previously cited cross-sectional study observed a decrease in the tumor-necrosis-factor α (TNF-α) plasma level in CUD patients assessed during abstinence compared with healthy control subjects [64]. Furthermore, higher TNF-α levels were associated with more severe withdrawal symptoms during inpatient cessation treatment [66]. 

Among the other potential inflammatory biomarkers, we identified two studies that assessed morning salivary cortisol, the stress hormone secreted by the adrenal glands following pituitary adreno corticotropic hormone (ACTH) stimulation. Ligabue et al. observed that increased cortisol levels at inpatient treatment entry were predictive of early treatment dropout before the scheduled 14 days of inpatient stay [68]. Furthermore, Sampedro-Piquero et al. showed that a higher salivary morning cortisol concentration was associated with CUD severity factors (duration of the disorder and craving scores) [69]. 

For the biomarkers of oxidative stress and inflammation, a lower level of plasmatic kynurenic acid, which is a peripheral marker of CNS inflammation and stress, was observed in a cross-sectional study comparing CUD patients in treatment with healthy controls [65]. Furthermore, a higher thiobarbituric acid reactive substances (TBRAS) level, which reflects lipid peroxidation, was associated with more severe CUD [58]. 

Finally, another study focused on glutathione, a peptide with antioxidant properties. The authors observed an increase in the plasmatic glutathione level between entry and day 14 of an inpatient cocaine cessation period [55]. Furthermore, there was a positive correlation between the duration of CUD prior to entry and the observed increase in glutathione. 

### 3.4. “Other” Biomarkers

Three other articles not included in the above categories but still relevant to the study of biomarkers related to abstinence maintenance were classified in the “other biomarkers” category (See Table 6). 

One investigated the plasmatic prolactin level. Prolactin is a peptidic hormone secreted by the anterior part of the pituitary gland with multiple roles, including in lactation. However, it is noticeable that dopamine serves as an inhibitor of pituitary prolactin secretion. Thus, the plasmatic prolactin level can be, among other causes, provoked by the increased dopaminergic transmission that is observed in CUD subjects. In Patkar et al., 2002 study, higher prolactin levels were observed in CUD compared with healthy control subjects, and this was associated with lower abstinence and response to treatment (evaluated by the counselors) in a 2-week outpatient follow-up period [73].

The two other biomarkers are related to the nutritional status of CUD patients, who often display weight loss due to increased motor activity and higher metabolism during periods of cocaine use combined with a loss of appetite. First, increased CUD severity has been associated with decreased leptin levels [71]. Finally, more severe withdrawal symptoms have been correlated with an increased level of plasma adipokines, which also have pro-inflammatory properties [72]. These results are in line with the expected relationship between the severity of CUD and weight loss, predicting relapse. 

## 4. Discussion

The most extensive body of literature on candidate biomarkers of relapse in CUD includes studies investigating functional neuroimaging markers during specific tasks.

One of the main results of these studies is the role of the striatum in determining the severity of CUD and abstinence. The results particularly highlight the role of the ventral striatum and, therefore, the nucleus accumbens, whose activity has been associated with a decrease in abstinence in all studies [18,26,32]. Beyond CUD, the NAcc has already been shown to be involved in addictions like alcohol use disorder [74], and in several psychiatric diseases such as depression and obsessive-compulsive disorder. The NAcc is highly innervated in dopaminergic projections and plays a major role in the reward system, which is well known to be involved in the development and maintenance of addictions in humans [75]. Furthermore, a study published in 2020 by Lee et al. suggested that the NAcc could be involved in susceptibility to cocaine-seeking behavior in rodents by acting on the cholinergic interneurons [76]. The striatum therefore seems to be a target of interest for the study of biomarkers of relapse in CUD. 

Another structure that was investigated in several studies included in this review is the thalamus. Results regarding the role of the thalamus in abstinence have been more heterogeneous: some studies have linked its decreased functioning to a higher abstinence maintenance rate [30,32], while others have associated it with worse abstinence maintenance [33] through various neuropsychological challenges. An article published by Huang et al. in 2018 shows that the thalamus seems to be involved in drug-seeking behaviors in rodents [77]. By extending its role to humans, the activation pattern of the thalamus when reacting to drug cues could be associated with cue sensitivity and, therefore, with the severity and the relapse of substance use disorder. The thalamus is, therefore, another structure of choice for the development of future neuroimaging studies on brain markers of CUD and abstinence maintenance. All studies presented in this review showing the role of the thalamus in CUD were conducted on a small number of patients (between fifteen and twenty). Replication studies with larger sample sizes and with cocaine-cue-induced tasks are needed to confirm the role of this region in CUD. 

Studies using other imaging techniques are less numerous. Neuroimaging of addictions mostly uses the fMRI technique, which allows neuronal activity to be captured during tasks assessing, for example, craving, flexibility, or inhibition. Indeed, neuropsychological tasks, such as delay discounting tasks or the Stroop task, require the assessment of cognitive performances relevant to CUD, such as inhibition processes and the activation of specific brain networks involved in CUD. One study used the same type of functional recording of the brain, real-time EEG recording, during a cue-induced craving task to assess salience and attention to visual cues [47]. The reaching of a consensus among different research teams to use the same experimental tasks to assess relevant brain functions could improve relapse prediction. 

Resting-state studies have investigated many structures and brain regions associated with different functionally related networks, making it difficult to find consistent results among resting-state studies, as the networks studied differ. 

The only DTI study included in this review associated a better white matter integrity in several brain structures such as the frontal, parietal, temporal, and occipital lobes with better abstinence [41]. This is in line with other studies on substance abuse where white matter integrity was impaired, but the link with relapse was not investigated [78]. It would therefore be interesting to study white matter integrity with DTI neuroimaging in prospective studies to test the association with relapse prediction. 

Other studies that used neuroimaging (anatomical MRI, PET-SCAN) did not identify biomarkers that may be considered potential biomarkers of relapse. 

Regarding studies assessing neuropsychology and abstinence, the Stroop task remains the reference test, despite its known limitations [79]. Of the four studies using the Stroop task, only one used a cocaine-word task, which allowed the attentional bias toward cocaine to be studied [49]. The results of these four studies are divergent and do not allow us to identify the Stroop task as a potential candidate biomarker to predict relapse. These observations are consistent with those of presented in a review published in 2015 by Christiansen et al., which studied the clinical relevance of the Stroop task and attentional bias in addiction [80]. They concluded that the predictive validity of attentional bias toward relapse is questionable, in particular because of the environment and internal factors involved. Here, again, a consensus on the most appropriate methods of assessment by research teams would help to achieve more consistent results. 

For neurotrophins, the literature relies mainly on BDNF, but highly divergent results have been obtained. Indeed, some studies associated increased BDNF levels with abstinence maintenance, and others associated it with relapse. Other studies regarding the role of BDNF in addictions and drug reward concluded that the role of this growth factor may differ according to the disorder stage and even according to sex [81,82].

Studies concerning genetics have mainly been carried out on animal models and were therefore excluded from our research. There have been few studies on the genetics of CUD in humans and, in particular, on relapse, mainly because genetic vulnerability is considered a stable trait, although that may change in the future with the development of epigenetic or transcriptomics studies. These techniques are now available for use in preclinical studies and focus on several rat brain sub-regions but may soon have the capacity for use in human blood samples. Ideally, once transcriptomic biomarkers of striatum activity are repeatedly available in the peripheral blood of human subjects, they should be combined with fMRI studies visualizing striatum activity during cue-induced or inhibition challenges. 

Concerning the immune system, in addition to their primary roles in mediating neuroinflammation, neuroimmune factors, such as cytokines and chemokines, are essential for various physiological brain functions. Expressed in neurons and glia [83], these molecules regulate synaptic function, mediate neuron–glia communication, interact with neuroendocrine and neuropeptide systems, and regulate neurogenesis and CNS development.

The role of the immune system has already been studied in relation to addiction. Indeed, studies on patients with alcohol use disorder have demonstrated a positive correlation between craving for alcohol and levels of pro-inflammatory cytokines (such as TNF-α and IL-6). The authors even suggested that activation of the immune system could increase cravings and, therefore, alcohol consumption [84]. 

Araos et al. demonstrated decreased plasma TNF-α levels among abstinent crack users compared with non-abstinent users. However, this study was not conducted with the hypothesis of finding a relationship between the maintenance of abstinence and plasmatic concentrations of cytokines, so in this article, TNF-α was used as a “state” marker [64]. Additionally, increased TNF-α levels have been associated with more severe withdrawal symptoms [66].

Repeated cocaine use, in addition to the direct pharmacological action on the dopamine transporter, is capable of activating the immune system in the CNS and probably in other organs such as the liver, kidneys, and heart. This, therefore, could cause cellular damage, especially related to oxidative stress processes. In the brain, inflammation caused by cocaine through these oxidative stress reactions could also alter the hypothalamic–pituitary–adrenal axis (HPA axis) [85]. Indeed, the modification of cytokine production may be linked to an alteration of the HPA axis, particularly regarding cortisol levels. 

Ligabue et al. highlighted a negative relationship between the salivary cortisol concentration and retention in a withdrawal program in cocaine users [68], suggesting a dysregulation of the HPA axis. Furthermore, a significant positive correlation between morning cortisol levels and craving levels in abstinent cocaine users has been demonstrated [69]. Additionally, the cortisol level could be an interesting biomarker to predict the risk of relapse in CUD patients and in cocaine users, as salivary cortisol would probably be easier to measure than plasma cytokines. It would also be interesting to further explore cortisol measures and more general stress-related factors combined with neuropsychological measures of attentional bias and/or inhibition capacities and brain function. 

Finally, concerning “other” biomarkers, one of the significant results regarding relapse was higher prolactin plasmatic levels, which were associated with poorer treatment responses in a prospective study of 86 cocaine-dependent subjects [73]. Additionally, Levandowski et al. associated greater withdrawal symptoms during crack-cocaine abstinence with an increased plasmatic level of adipokines, which are linked to an inflammatory state [72].

Furthermore, several studies included in this review, mostly concerning the immune system and oxidative stress, explored the mechanisms of CUD rather than identifying predictors of abstinence. As most of these studies were cross-sectional association studies, and because prospective studies are mostly limited to neuroimaging and neuropsychological testing and some plasmatic BDNF studies, the replication of the observed biomarkers in independent samples with a hypothesis-free design is lacking. Even in prospective studies, the design, data collection, and statistical analysis allow correlations rather than conclusions about biomarkers to be made. Mechanistic studies in animal models would be a step forward in the validation of those correlations. Identifying and validating biomarkers of relapse in cocaine use disorder would allow the development of personalized medicines. Patients’ treatment could also be matched with biomarkers of treatment response, and patients displaying biomarkers associated with a higher risk of relapse could be offered more intensive care. As the field of biomarkers of relapse in CUD is rapidly developing, future studies must now overcome the main pitfalls identified. Until now, authors have usually chosen to assess one biomarker at a time, apart from fMRI studies, which combine striatum neuroimaging and neuropsychology. There is a gap to be filled in the next few years. Future studies should integrate peripheral biomarkers and brain functioning biomarkers. Brain-derived or inflammatory biomarkers assessed in blood samples should now be combined with brain functioning assessments in multimodal prediction models. In particular, the combination of real-time repetitive time-sensitive blood sampling and functional brain imagery during neuropsychological challenges, assessing both the attentional bias toward cocaine and its counterpart, the inhibitory capacity, to prevent cocaine use resuming despite such a bias, is recommended. Replication studies using independent samples and a prospective design could validate the predictive value of the identified combination of biomarkers. 

## 5. Conclusions

Despite the diversity of the literature on brain and biological changes related to chronic cocaine use, few studies have identified validated biomarkers of relapse. Most of the initial studies in this area were cross-sectional, and the field has recently moved to prospective designs. The best candidate predictors have emerged from fMRI studies investigating the structure and function of the reward system, especially the striatum, during neuropsychological challenges. Studies using isolated neuropsychological challenges of attentional bias have also been linked with relapse. To validate these candidate biomarkers, prospective studies on independent samples, combining different approaches, are warranted to confirm their association with relapse and the relevance of their predictive value.

## Figures and Tables

**Figure 1 brainsci-12-01013-f001:**
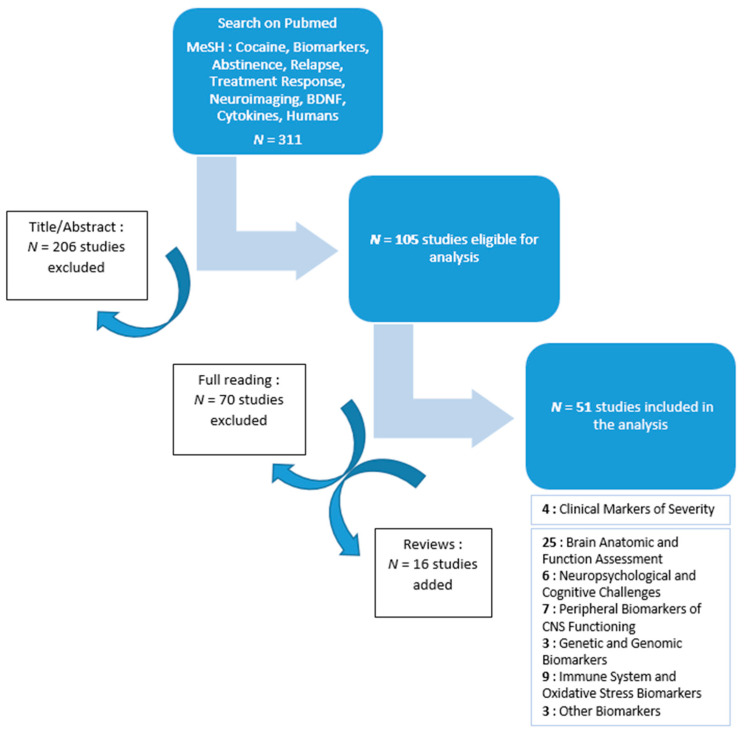
Flow-chart of the article selection process.

**Table 1 brainsci-12-01013-t001:** Brain anatomic and function assessment.

	Participants	Study Design	Article
Task-based fMRI:			
Biomarker of:			
↗ abstinence: ↗ midbrain activity during a MIDT ↘ abstinence: ↗ ventral striatum and NAcc activity during a MIDT ↘ craving: ↗ ventral striatum activity during a MIDT	*N* = 57	Prospective	Balodis et al., 2016 [26]
↗ cue-induced craving: ↗ right ventral striatum activity during a CCQ-N	*N* = 20	Cross-sectional	Bell et al., 2014 [18]
↗ abstinence: ↘ dlPFC and left PCC activity during a Stroop Task ↘ abstinence: ↘ right striatum activity during a Stroop Task	*N* = 22	Prospective	Brewer et al., 2008 [27]
↗ abstinence: ↗ PCC activity during an Odball Task	*N* = 45	Prospective	Clark et al., 2014 [20]
↘ lifetime stimulant use: ↘ dACC activity during a reaction time task	*N* = 87	Cross-sectional	Claus et al., 2018 [28]
↗ abstinence: ↗ PFC activity during a response inhibition task	*N* = 27	Cross-sectional	Connolly et al., 2012 [29]
↘ abstinence: ↗ thalamus, right caudate and culmen activity during a MIDT ↘ treatment retention: ↗ left amygdala and left PHG activity during a MIDT	*N* = 20	Prospective	Jia et al., 2011 [30]
↗ relapse: ↗ PrG, PCC, STG, LiG, and IOG activity during exposure to cocaine-related cues	*N* = 17	Prospective	Kosten et al., 2006 [31]
↗ abstinence: ↘ intrinsic connectivity in ventral striatum, right insula, left hippocampus, substantia nigra, and thalamus during a Stroop Task	*N* = 16	Prospective	Mitchell et al., 2013 [32]
↘ abstinence: ↘ thalamus activity during a working memory task	*N* = 19	Cross-sectional	Moeller et al., 2010 [33]
↘ abstinence: ↗ right putamen, insula, and bilateral occipital regions during exposure to cocaine-related cues	*N* = 30	Prospective	Prisciandaro et al., 2013 [34]
↗ abstinence: ↘ amygdala, FuG and PHG activity during exposure to cocaine-related cues	*N* = 73	Prospective	Regier et al., 2021 [35]
↗ cocaine use in the prior month: ↘ thalamus and vmPFC during a reaction time task	*N* = 54	Cross-sectional	Zhang et al., 2014 [36]
Resting-state:			
Biomarker of:			
↗ relapse: ↗ rsFC between PCC and precuneus	*N* = 40	Prospective	Adinoff et al., 2015 [37]
↗ relapse: ↗ rsFC between NAcc and FPC, ↗ rsFC between NAcc and posterior cingulate gyrus, ↗ rsFC between subgenual ACC and left PFC	*N* = 18	Prospective	Camchong et al., 2014 [38]
↗ years of cocaine use: ↘ rsFC between VTA and thalamus, lentiform nucleus and NAcc	*N* = 39	Cross-sectional	Gu et al., 2010 [39]
↗ relapse: ↘ rsFC between left corticomedial amygdala and vmPFC and rACC ↗ abstinence: ↘ rsFC between bilateral amygdala and lingual gyrus and cuneus	*N* = 45	Prospective	McHugh et al., 2014 [19]
↘ relapse: ↗ rsFC in the executive control network, ↗ rsFC between bilateral dlPFC, IFG, right IPL, dmPFC, and left vlPFC ↗ relapse: ↗ rsFC in the default mode network, ↗ rsFC between bilateral precuneus and amygdala and hippocampus, ↗ rsFC between bilateral cuneus, visual cortex, vmOFC, thalamus, FuG, and left precuneus	*N* = 43	Prospective	Zhai et al., 2021 [40]
Diffusion Tensor Imaging (DTI):			
Biomarker of:			
↗ abstinence: ↗ WM integrity in frontal, parietal, and occipital lobes and rostral corpus callosum, cerebellum, and rostral midbrain	*N* = 16	Prospective	Xu et al., 2010 [41]
Positron Emission Tomography (PET-SCAN):			
Biomarker of:			
↗ abstinence: ↗ creatine levels in frontal and parietal GM and frontal WM, ↗ choline-containing metabolites levels in parietal GM, ↗ myo-inositol levels in thalamus	*N* = 18	Cross-sectional	Abé et al., 2013 [42]
↗ cocaine use: ↘ GABA levels in ACC, N-acetylaspartate levels in cortical GM	*N* = 28	Cross-sectional	Abé et al., 2013 [43]
↗ abstinence: ↗ dopamine signaling in the limbic striatum	*N* = 25	Prospective	Martinez et al., 2011 [44]
Anatomical MRI:			
Biomarker of:			
↗ years of cocaine use: ↘ GM volume in cingulate cortex and bilateral superior frontal gyri	*N* = 84	Cross-sectional	Ide et al., 2014 [45]
↘ abstinence: ↗ WM volume in hippocampus	*N* = 23	Prospective	Xu et al., 2014 [46]
Electroencephalography:			
Biomarker of:			
↗ craving: ↗ cue-induced LPP at 1 and 6 months of abstinence	*N* = 76	Cross-sectional	Parvaz et al., 2016 [47]

↘: decreased, ↗: increased, ACC: anterior cingulate cortex, CCQ-N: Cocaine Craving Questionnaire-Now, dACC: dorsal anterior cingulate cortex, dlPFC: dorsolateral prefrontal cortex, dmPFC: dorsomedial prefrontal cortex, FuG: fusiform gyrus, FPC: frontopolar cortex, GM: gray matter, IFG: inferior frontal gyrus, IOG: inferior occipital gyrus, IPL: inferior parietal lobule, LiG: lingual gyrus, LPP: late positive potential, MIDT: Monetary Incentive Delay Test, NAcc: nucleus accumbens, PCC: posterior cingulate cortex, PFC: prefrontal cortex, PHC: parahippocampal gyrus, PrG: precentral gyrus, rACC: rostral anterior cingulate cortex, rsFC: resting-state functional connectivity, STG: superior temporal gyrus, vlPFC: ventrolateral prefrontal cortex, vmOFC: ventromedial orbitofrontal cortex, vmPFC: ventromedial prefrontal cortex, VTA: ventral tegmental area, WM: white matter.

**Table 2 brainsci-12-01013-t002:** Neuropsychological and cognitive challenges.

	Participants	Study Design	Article
Biomarker of:			
↗ cocaine reactivity: ↗ attentional bias during a cocaine-word Stroop Task	*N* = 114	Cross-sectional	Anastasio et al., 2014 [49]
↗ treatment retention: ↘ Stroop interference effect	*N* = 22	Prospective	Brewer et al., 2008 [32]
↗ relapse: ↘ cognitive flexibility scores on the WCST	*N* = 45	Prospective	Clark et al., 2014 [20]
↗ relapse: ↘ memory score on the RAVLT	*N* = 36	Prospective	Fox et al., 2009 [50]
↗ abstinence: ↗ Stroop interference effect	*N* = 13	Prospective	Mitchell et al., 2013 [32]
↗ treatment dropout: ↗ attentional bias during a Stroop Task	*N* = 74	Prospective	Streeter et al., 2008 [51]

↘: decreased, ↗: increased, RAVLT: Rey Auditory Verbal Learning Test, WCST: Wisconsin Card Sorting Test.

**Table 3 brainsci-12-01013-t003:** Peripheral biomarkers of CNS function.

	Participants	Study design	Article
Biomarker of:			
↗ abstinence: ↘ BDNF levels	*N* = 40	Prospective	Corominas-Roso et al., 2015 [15]
↘ abstinence: ↗ BDNF levels	*N* = 35	Prospective	D’Sa et al., 2011 [53]
↗ abstinence: ↗ BDNF levels	*N* = 22	Cross-sectional	Hilburn et al., 2011 [54]
↗ abstinence: ↘ BDNF levels between entry and 2 weeks of detoxification	*N* = 31	Prospective	Hirsch et al., 2018 [55]
↗ severity of CUD: ↗ n-methylserotonin levels, ↘ xanthine levels	*N* = 18	Cross-sectional	Patkar et al., 2009 [56]
↗ abstinence: ↗ BDNF levels	*N* = 47	Prospective	Scherer et al., 2016 [57]
↗ severity of CUD: ↘ BDNF levels	*N* = 49	Prospective	Sordi et al., 2014 [58]

↘: decreased, ↗: increased, BDNF: Brain-Derived Neurotrophic Factor, CUD: cocaine use disorder.

**Table 4 brainsci-12-01013-t004:** Genetic and genomic biomarkers.

	Participants	Study Design	Article
Biomarker of:			
↗ cocaine cue reactivity: carrying the SER23 variation of the 5-HT2C receptor	*N* = 114	Cross-sectional	Anastasio et al., 2014 [49]
↘ time before relapse: variation in the βnAC subunit that possibly alters CHRNA5 gene expression	*N* = 581	Cross-sectional	Forget et al., 2021 [60]
↗ severity of CUD: upregulation of miR-181	*N* = 30	Cross-sectional	Viola et al., 2019 [61]

↘: decreased, ↗: increased, 5-HT2C receptor: 5-hydroxytryptamine receptor 2C, CHRNA5: cholinergic receptor nicotinic alpha5 subunit, miR: microRNA, nAChR: nicotinic acetylcholine receptor.

**Table 5 brainsci-12-01013-t005:** Immune system and oxidative stress biomarkers.

	Participants	Study Design	Article
Biomarker of:			
↗ abstinence: ↘ TNF-α, MCP-1 and SDF-1 levels ↗ severity of CUD: ↗ IL-1β, fraktaline and SDF-1 levels	*N* = 82	Cross-sectional	Araos et al., 2015 [64]
↗ lifetime dependence: ↘ kynurenic acid levels	*N* = 100	Cross-sectional	Araos et al., 2019 [65]
↗ CUD duration: ↗ glutathione levels between entry and 3 weeks of detoxification	*N* = 31	Prospective	Hirsch et al., 2018 [55]
↗ abstinence symptoms severity: ↗ TNF-α levels	*N* = 44	Prospective	Levandowski et al., 2014 [66]
↘ cognitive flexibility: ↗ IL-6 levels	*N* = 42	Cross-sectional	Levandowski et al., 2016 [67]
↘ treatment retention: ↗ entry cortisol levels	*N* = 44	Prospective	Ligabue et al., 2020 [68]
↗ craving: ↗ morning cortisol levels	*N* = 14	Cross-sectional	Sampedro-Piquero et al., 2020 [69]
↗ severity of CUD: ↗ TBARS levels (lipid oxidation)	*N* = 49	Prospective	Sordi et al., 2014 [58]
↗ years of cocaine use: ↘ IL-29 and IL-20 levels	*N* = 85	Cross-sectional	Stamatovich et al., 2021 [70]

↘: decreased, ↗: increased, IL: Interleukin, MCP-1: Monocyte chemoattractant protein-1, SDF-1: stromal cell-derived factor 1, TBARS: Thiobarbituric Acid Reactive Substances, TNF-α: tumor necrosis factor.

**Table 6 brainsci-12-01013-t006:** “Other” biomarkers.

	Participants	Study Design	Article
Biomarker of:			
↗ severity of CUD: ↘ leptin plasmatic levels	*N* = 40	Cross-sectional	Escobar et al., 2018 [71]
↗ withdrawal symptoms: ↗ plasmatic adipokine levels	*N* = 104	Cross-sectional	Levandowski et al., 2013 [72]
↘ abstinence: ↗ prolactin plasmatic levels	*N* = 86	Prospective	Patkar et al., 2002 [73]

↘: decreased, ↗: increased.

## Data Availability

Not applicable.

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
