# Peer review of "Biomarkers of Relapse in Cocaine Use Disorder: A Narrative Review"

_brainsci, 2022, doi:10.3390/brainsci12081013_

Round 1

Reviewer 1 Report

The manuscript by Poireau and colleagues is a review of published clinical literature focused on biomarkers associated with relapse risk in individuals with cocaine use disorder. I found the review informative and found the information in the tables generally well-organized and easy to follow. While the grammar/sentence structure needs to be edited some, the scientific summary was generally easy to follow. I did notice that there is a paper that was not included in the review that I believe should be included. Please review the following paper and include in the appropriate Table/text discussion:

Parvaz MA, Moeller SJ, Goldstein RZ (2016). Incubation of Cue-Induced Craving in Adults Addicted to Cocaine Measured by Electroencephalography. JAMA Psychiatry 73: 1127-1134. PMID 27603142.  

Reviewer 2 Report

In the reviewed manuscript „Biomarkers of relapse in cocaine use disorder: a narrative review”, the authors reviewed the available literature from the last 20 years, concerning the potential biomarkers associated with relapse in cocaine use disorder. This is an important issue due to the increasing number of people with the problem of substance use disorder and the lack of effective therapy. Before being accepted for publication, it is worth considering some additions and correcting a few editorial mistakes.

1. In my opinion, it is worth considering supplementing the manuscript with information on the features that a biomarker should fulfill for the described disorder so that it can be effectively and commonly used in clinical practice.

2. The discussion and summary also lack the in-depth analysis of the authors indicating the most promising path for the search and validation of the biomarker associated with relapse in cocaine use disorder as well as the challenges and threats that may arise during the research.

3. It is worth considering listing examples of external stimulus in line 60 (page 2), which will be helpful in understanding the text for a wider group of readers.

4. In Figure 1, the sum total of the publications mentioned in each category is greater (56) than the n = 50 indicated in the blue box. Please check this.

5. The format of the references in the text and in the list of references should be carefully checked and standardized in accordance with the requirements of the journal.

6. Readers will find it helpful if there is a list of the explained abbreviations and symbols used in each of them below the tables. Additionally, the positioning of the text in the tables should be corrected, which in some places makes it difficult to use.

7. Please pay attention to editorial errors, eg missing '.' at the end of the sentence on lines 154 and 171 (page 5), and 213 (page 9).

8. No reference at the end of a sentence on line 52 (page 2).

9. Please add the abbreviations of CNS and TNF-alpha expansion (and use the unified version of alpha or a) when first used in the text and then use only the abbreviation.

10. Line 279 (page 12) (Lewandowski et al., 2014) - no reference in the literature list (perhaps it is a typo 'w' instead of 'v'). Please correct.

Reviewer 3 Report

The text is interesting in that it provides a comprehensive summary of studies conducted in the cocaine use disorder population. I find it unbalanced that there is a specific section on BDNF with 10 articles and another 10 articles on other markers and it is not well defined why.

The study selection and methodology does not specify how many studies are included and why these keywords are selected; the inclusion and exclusion criteria of the studies should be better defined for a better understanding.

Round 2

Reviewer 2 Report

The authors have adequately addressed my initial concerns about the manuscript. Thank you for your work in clarifying these points. Minor comments to the revision version of the manuscript Biomarkers of relapse in cocaine use disorder: a narrative review:
1. Table 3 (Page 9, Line 262): Please correct the abbreviation of BDNF.
2. Page 12, Line 390: Please correct the format of references (Parvaz et al 2016) to [44].

This manuscript is a resubmission of an earlier submission. The following is a list of the peer review reports and author responses from that submission.